# Influence of nutritional supplements on antibody levels in pregnant women vaccinated with inactivated SARS-CoV-2 vaccines

**Xi Zhang[1], Xue Han[1], Baolan Chen[1], Xi Fu[1], Yajie Gong[2]\*, Wenhan Yang[3]\*, Qingsong Chen****[1]\***

1 Department of Occupational and Environmental Health, School of Public Health, Guangdong Pharmaceutical University, Guangzhou, Guangdong Province, China, 2 Department of Epidemiology and Medical Statistics, School of Public Health, Guangdong Pharmaceutical University, Guangzhou, Guangdong Province, China, 3 Department of Child and Adolescent Health, School of Public Health, Guangdong Pharmaceutical University, Guangzhou, Guangdong Province, China

\* gongyajie2016@163.com (YG); wenhan-yang@gdpu.edu.cn (WY); qingsongchen@aliyun.com (QC)

## Abstract

### Background

Because of the significantly higher demand for nutrients during pregnancy, pregnant women are more likely to have nutrient deficiencies, which may adversely affect maternal and fetal health. The influence of nutritional supplements on the immune effects of inactivated SARS-CoV-2 vaccines during pregnancy is not clear.

### Methods

In a multicenter cross-sectional study, we enrolled 873 pregnant women aged 18–45 y in Guangdong, China. The general demographic characteristics of pregnant women and their use of nutritional supplements were investigated, and the serum antibody levels induced by inactivated SARS-CoV-2 vaccines were measured. A logistic regression model was used to analyze the association between nutritional supplements and SARS-CoV-2 antibody levels.

### Results

Of the 873 pregnant women enrolled, 825 (94.5%) took folic acid during pregnancy, 165 (18.9%) took iron supplements, and 197 (22.6%) took DHA. All pregnant women received at least one dose of inactivated SARS-CoV-2 vaccine, and the positive rates of serum SARS-CoV-2 neutralizing antibodies (NAbs) and immunoglobulin G (IgG) antibodies were 44.7% and 46.4%, respectively. After adjustment for confounding factors, whether pregnant women took folic acid, iron supplements, or DHA did not influence NAb positivity or IgG positivity ($P > 0.05$). Compared with pregnant women who did not take folic acid, the odds ratios (ORs) for the presence of SARS-CoV-2 NAb and IgG antibody in pregnant women who took folic acid were 0.67 ($P = 0.255$; 95% CI, 0.34–1.32) and 1.24 ($P = 0.547$; 95% CI, 0.60–2.55), respectively. Compared with pregnant women who did not take iron supplements, the

---

**Data Availability Statement:** All relevant data are within the manuscript and its Supporting information files.

**Funding:** This study was funded by the Exposure to SARS-CoV-2 vaccine before or during pregnancy and adverse pregnancy outcomes: a cohort study grant [41-43241529], the 2022 Science and Technology Innovation Project of Guangdong Medical Products Administration "Discussion on active monitoring methods for adverse reactions of vaccines on the market based on the real world" grant [2022TDZ21], and by the 2022 Science and Technology Innovation Project of Guangdong Medical Products Administration "Research and application of key technology and evaluation system of pharmacovigilance" project funding [2022ZDZ06]. The funders had no role in study design, data collection and analysis, decision to publish, or preparation of the manuscript.

**Competing interests:** The authors have declared that no competing interests exist.

ORs for the presence of NAb and IgG antibody in pregnant women who took iron supplements were 1.16 ($P$ = 0.465; 95% CI, 0.77–1.76) and 0.98 ($P$ = 0.931; 95% CI, 0.64–1.49), respectively. Similarly, the ORs for NAb and IgG antibody were 0.71 ($P$ = 0.085; 95% CI, 0.49–1.04) and 0.95 ($P$ = 0.801; 95% CI, 0.65–1.38) in pregnant women who took DHA compared with those who did not.

## Conclusions

Nutritional supplementation with folic acid, iron, or DHA during pregnancy was not associated with antibody levels in pregnant women who received inactivated SARS-CoV-2 vaccines.

## 1 Introduction

Since the outbreak of COVID-19, research and development of vaccines and various antiviral drugs to combat COVID-19 and its complications have increased worldwide [1–4]. At present, dozens of COVID-19 vaccines have been approved for use around the world. Because pregnant women have not been included in any clinical trials of COVID-19 vaccines [5, 6], data on the efficacy and safety of these vaccines for pregnant women are limited. In addition, pregnant women have a significantly higher risk of COVID-19 infection, severe disease, and death due to changes in their immune response during pregnancy [7–9]. Therefore, it is necessary to understand the risks faced by pregnant women and the associated protective factors.

Adequate nutrition is essential to ensure the development, operation, and maintenance of the immune system, and nutritional supplements play an important role [10–12]. When malnutrition occurs, the immune response is affected, making the body susceptible to infection, which in turn exacerbates malnutrition. Excessive nutrient intake can adversely affect on all components of the immune system. Therefore, a well-balanced diet that includes plenty of nutrient-rich foods and supplements is necessary to prevent infectious diseases and ensure optimal immune function [12, 13]. Some studies have shown that nutrient supplements are particularly important during pregnancy. For example, folic acid is an indispensable nutrient during pregnancy, crucial in preventing neural tube defects [14]. In addition, folic acid supplementation during pregnancy has been found to reduce the risk of preeclampsia and preterm birth [15].

Iron supplements are also necessary during pregnancy. Iron is an essential trace element for the human body. It supports hemoglobin synthesis and is used in the formation of iron-containing enzymes. It also constitutes the body's iron storage in the form of ferritin [16, 17]. Iron deficiency anemia is one of the most common complications of pregnancy and a risk factor for excessive bleeding during childbirth, premature birth, and low birth weight [18, 19]. Pregnant women with iron deficiency anemia are more susceptible to SARS-CoV-2 infection [20].

DHA is an important component of brain cell membranes and is essential for fetal brain development. DHA supplementation during pregnancy promotes fetal brain development and plays an important role in increasing fetal intelligence and improving vision [21, 22]. The World Health Organization recommends that pregnant women should supplement with at least 300mg of DHA per day. Because of its immunomodulatory effects, DHA supplementation during pregnancy can also improve the immune health of the fetus.

Nutritional status during pregnancy is crucial. Many nutrients have potent immunomodulatory effects that can modify susceptibility to COVID-19 infection. It is worth exploring

whether nutritional supplements can increase the immune response to COVID-19 vaccines. Therefore, the purpose of this study was to investigate the effect of dietary supplements on antibody levels in pregnant women vaccinated with inactivated SARS-CoV-2 vaccine. This study can provide a certain reference for the question of the influence of women taking nutrients during pregnancy on the level of antibody produced by COVID-19 vaccines.

## 2 Materials and methods

### 2.1 Study design and inclusion criteria

This multicenter study was performed in the obstetric clinics of three hospitals in Guangdong Province, China from December 2021 to July 2022. Inclusion criteria included (1) receiving at least one dose of inactivated vaccine before or during pregnancy, (2) being pregnant and having not been infected with COVID-19, and (3) agreeing to participate and providing informed consent. Pregnant women who were infected with COVID-19 after enrollment and were vaccinated with noninactivated COVID-19 vaccines were excluded. A total of 873 pregnant women met the inclusion criteria (Fig 1). The study was approved by the Medical Ethics Review Board of the School of Public Health, Guangdong Pharmaceutical University (IRB 2021–01), and complied with the Declaration of Helsinki guidelines. All participants signed paper informed consent form and minors were excluded from this study.

### 2.2 Study variable

We developed a survey questionnaire for pregnant women that included questions such as age (stratified by advanced maternal age as defined by the World Health Organization), body mass index(BMI), gravidity and parity, smoking, doses of vaccine taken, adverse events after vaccination, time since vaccination (defined as the time interval between the last dose of vaccine and blood collection), and other general demographic information, vaccination-related information, and information on the types of nutritional supplements used. Vaccination information was collected through the vaccination registration platform. The presence of nutrient supplements was the exposure variable, negative or positive antibody level were outcome variables, and age, BMI, parity, education and income level, smoking, number of exercises per week, doses of vaccination, adverse reactions, and time since vaccination were covariates.

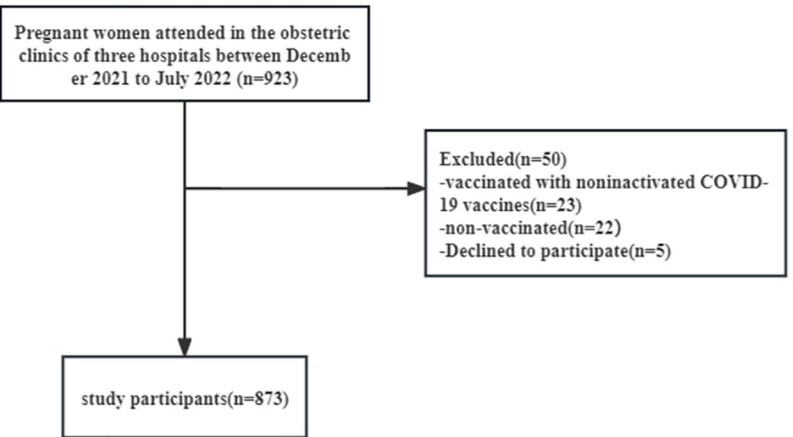

**Fig 1. Flow chart of study participants.**

## 2.3 Sample collection and analysis

After enrollment, 5 mL of venous blood was collected, centrifuged to separate the serum, and stored in a –80˚C freezer. Neutralizing antibody (NAb) levels were determined via a cytopathogenic effect assay. Specifically, serum samples were inactivated at 56˚C for 30 min, diluted fourfold, and incubated with an equal volume (50 μL) of live SARS-CoV-2 virus suspension at 36.5˚C in 5% $CO_2$ for 2 h. Vero cells ($1.0-2.0 \times 10^5$ cells/mL) were then added to the serum–virus suspension in the microplates and incubated in duplicate at 36.5˚C in 5% $CO_2$ for 5 d. Cytopathological effects were observed under the microscope. A NAb titer of <1:8 was considered negative and ≥1:8 was considered positive. SARS-CoV-2 immunoglobulin G (IgG) antibody was detected by chemiluminescence (Bioscience, Chongqing, China). An antibody level of <1.00 signal to cutoff ratio was considered negative, and a level of ≥1.00 was considered positive.

## 2.4 Statistical analysis

Statistical analysis was performed in IBM SPSS version 25 (SPSS Inc., IBM Corp., Armonk, NY). We analyzed demographic data, vaccination-related information, and antibody-positive rates of pregnant women according to whether they were taking nutritional supplements. Chi-squared and Fisher tests were used to compare proportions, when the sample size n < 40 or the expected frequency T < 1 use Fisher's test. Mann–Whitney U-test and Kruskal–Wallis test were used for nonnormal data. The logistic regression model was used to analyze the association between nutritional supplements and SARS-CoV-2 antibody positive rate. The adjusted factors were determined through the results of univariate analysis and literature review. Model 1 was adjusted for age, BMI, education, gravidity, parity, adverse events, inoculation durations, doses of vaccination. Model 2 was adjusted for age, BMI, smoking, number of exercises per week, adverse events, inoculation durations, doses of vaccination. Model 3 was adjusted for age, BMI, smoking, number of exercises per week, adverse events, inoculation durations, doses of vaccination. All P-values were two-sided, and differences with P-values <0.05 were considered statistically significant.

## 3 Results

### 3.1 Demographics of the study population

Of the 873 pregnant women included in the study, 825 (94.5%) took folic acid, 165 (18.9%) took iron supplements, and 197 (22.6%) took DHA during pregnancy. All pregnant women received at least one dose of inactivated SARS-CoV-2 vaccine, and the positive rates of serum SARS-CoV-2 NAbs and IgG antibodies were 44.7% and 46.4%, respectively. Of the women included in the study 87.9% were <35 years old. Basic demographic characteristics are presented in (Tables 1 and 2).

We found that there were statistically significant differences between groups of pregnant women who did or did not take folic acid in terms of gravidity (P = 0.002), parity (P = 0.012), and education (P = 0.008). The percentage of pregnant women taking folic acid was higher in those with fewer than two pregnancies (80.6%), ≤1 birth (87.6%), and junior college education (34.7%). There were statistically significant differences between groups of pregnant women who did or did not take iron supplements in terms of smoking (P = 0.008), weekly exercise frequency (P < 0.001), vaccine doses (P < 0.001), time since vaccination (P < 0.001), SARS-CoV-2 neutralizing antibody (P = 0.017), and IgG antibody (P < 0.001). The presence or absence of DHA supplementation was similar to the presence or absence of iron supplementation. There were significant differences between groups in smoking (P = 0.017), weekly

**Table 1. Demographic characteristics of the participants stratified by type of nutrient supplements.**

| Characteristics | Total (N = 873) | Folic acid | | P | Iron supplements | | P | DHA | | P |
|---|---|---|---|---|---|---|---|---|---|---|
| | | Take (N = 825) | No take (N = 48) | | Take (N = 165) | No take (N = 708) | | Take (N = 197) | No take (N = 676) | |
| **Age group, y** | | | | 0.054 | | | 0.270 | | | 0.117 |
| <35 | 768(87.9) | 730(88.5) | 38(79.2) | | 141(85.5) | 627(88.6) | | 167(84.8) | 601(88.9) | |
| ≥35 | 105(12.1) | 95(11.5) | 10(20.8) | | 24(14.5) | 81(11.4) | | 30(15.2) | 75(11.1) | |
| BMI[A] | | | | 0.118 | | | 0.846 | | | 0.832 |
| <18.5 | 168(19.2) | 160(19.4) | 8(16.7) | | 30(18.2) | 138(19.5) | | 35(17.8) | 133(19.7) | |
| 18.5–24 | 533(61.1) | 508(61.6) | 25(52.1) | | 104(63.0) | 429(60.6) | | 122(61.9) | 411(60.8) | |
| 24–28 | 137(15.7) | 128(15.5) | 9(18.7) | | 24(14.5) | 113(16.0) | | 31(15.7) | 106(15.7) | |
| ≥28 | 35(4.0) | 29(3.5) | 6(12.5) | | 7(4.2) | 28(3.9) | | 9(4.6) | 26(3.8) | |
| **Gravidity** | | | | 0.002* | | | 0.923 | | | 0.169 |
| 1 | 357(40.9) | 345(41.8) | 12(25.0) | | 68(41.2) | 289(40.8) | | 85(43.1) | 272(40.2) | |
| 2 | 337(38.6) | 320(38.8) | 17(35.4) | | 65(39.4) | 272(38.4) | | 81(41.1) | 256(37.9) | |
| ≥3 | 179(20.5) | 160(19.4) | 19(39.6) | | 32(19.4) | 147(20.8) | | 31(15.7) | 148(21.9) | |
| **Birth** | | | | 0.012* | | | 0.870 | | | 0.377 |
| 0 | 423(48.5) | 405(49.1) | 18(37.5) | | 77(46.7) | 346(48.9) | | 94(47.7) | 329(48.7) | |
| 1 | 335(38.4) | 318(38.5) | 17(35.4) | | 66(40.0) | 269(38.0) | | 82(41.6) | 253(37.4) | |
| ≥2 | 115(13.1) | 102(12.4) | 13(27.1) | | 22(13.3) | 93(13.1) | | 21(10.7) | 94(13.9) | |
| **Education** | | | | 0.008* | | | 0.381 | | | 0.077 |
| MIDDLE SCHOOL AND BELOW | 219(25.1) | 198(24.0) | 21(43.8) | | 40(24.2) | 179(25.3) | | 39(19.8) | 180(26.6) | |
| HIGH SCHOOL | 187(21.4) | 176(21.3) | 11(22.9) | | 38(23.0) | 149(21.0) | | 43(21.8) | 144(21.3) | |
| JUNIOR COLLEGE | 294(33.7) | 286(34.7) | 8(16.7) | | 48(29.1) | 246(34.7) | | 65(33.0) | 229(33.9) | |
| BACHELOR AND ABOVE | 173(19.8) | 165(20.0) | 8(16.7) | | 39(23.6) | 134(18.9) | | 50(25.4) | 123(18.2) | |
| **Household income per capita, m** | | | | 0.061 | | | 0.847 | | | 0.076 |
| <4000 | 139(15.9) | 129(15.6) | 10(20.8) | | 23(13.9) | 116(16.4) | | 23(11.7) | 116(17.2) | |
| 4000~6000 | 269(30.8) | 248(30.1) | 21(43.8) | | 50(30.3) | 219(30.9) | | 57(28.9) | 212(31.4) | |
| 6000~10000 | 298(34.1) | 285(34.5) | 13(27.1) | | 58(35.2) | 240(33.9) | | 69(35.0) | 229(33.9) | |
| ≥10000 | 167(19.2) | 163(19.8) | 4(8.3) | | 34(20.6) | 133(18.8) | | 48(24.4) | 119(17.6) | |
| **Smoking** | | | | 0.067 | | | 0.008* | | | 0.017* |
| NO | 865(99.1) | 819(99.3) | 46(95.8) | | 160(97.0) | 705(99.6) | | 192(97.5) | 673(99.6) | |
| YES | 8(0.9) | 6(0.7) | 2(4.2) | | 5(3.0) | 3(0.4) | | 5(2.5) | 3(0.4) | |
| **Number of exercises per week** | | | | 0.999 | | | <0.001* | | | 0.021* |
| 0 | 330(37.8) | 312(37.8) | 18(37.5) | | 39(23.6) | 291(41.1) | | 59(29.9) | 271(40.1) | |
| <3 | 55(6.3) | 52(6.3) | 3(6.3) | | 9(5.5) | 46(6.5) | | 11(5.6) | 44(6.5) | |
| ≥3 | 488(55.9) | 461(55.9) | 27(56.3) | | 117(70.9) | 371(52.4) | | 127(64.5) | 361(53.4) | |

[a]: Since the simple with BMI≥28 was small, it was combined with the simple with 24≤BMI<28 for statistical analysis.

*: $P < 0.05$

exercise frequency ($P = 0.021$), vaccine doses ($P < 0.001$), time since vaccination ($P = 0.001$), SARS-CoV-2 neutralizing antibody ($P < 0.001$), and IgG antibody ($P = 0.012$). There were no statistically significant differences in age, body mass index, per capita monthly income, or adverse events.

We further analyzed the general demographic characteristics of the pregnant women taking the three nutritional supplements separately. For pregnant women who took folic acid, there

**Table 2. Vaccination and SARS-CoV-2 antibody positivity rates stratified by type of nutrient supplements.**

| Characteristics | Total (N = 873) | Folic acid | | P | Iron supplements | | P | DHA | | P |
|---|---|---|---|---|---|---|---|---|---|---|
| | | Take (N = 825) | No take (N = 48) | | Take (N = 165) | No take (N = 708) | | Take (N = 197) | No take (N = 676) | |
| **Doses of vaccination** | | | | 0.901 | | | <0.001* | | | <0.001* |
| ONE DOSE | 84(9.6) | 79(9.6) | 5(10.4) | | 41(24.8) | 43(6.1) | | 30(15.2) | 54(8.0) | |
| TWO DOSES | 669(76.7) | 633(76.7) | 36(75.0) | | 114(69.1) | 555(78.4) | | 154(78.2) | 515(76.2) | |
| THREE DOSES | 120(13.7) | 113(13.7) | 7(14.6) | | 10(6.1) | 110(15.5) | | 13(6.6) | 107(15.8) | |
| **Adverse events** | | | | 0.207 | | | 0.621 | | | 0.825 |
| NO | 477(54.6) | 455(55.2) | 22(45.8) | | 93(56.4) | 384(54.2) | | 109(55.3) | 368(54.4) | |
| YES | 396(45.4) | 370(44.8) | 26(54.2) | | 72(43.6) | 324(45.8) | | 88(44.7) | 308(45.6) | |
| **Inoculation durations, wk** | | | | 0.096 | | | <0.001* | | | 0.001* |
| <20 | 119(13.6) | 109(13.2) | 10(20.8) | | 5(3.0) | 114(16.1) | | 8(4.1) | 111(16.4) | |
| 20–24 | 107(12.3) | 98(11.9) | 9(18.8) | | 10(6.1) | 97(13.7) | | 22(11.2) | 85(12.6) | |
| 24–28 | 202(23.1) | 188(22.8) | 14(29.2) | | 30(18.2) | 172(24.3) | | 50(25.4) | 152(22.5) | |
| 28–32 | 230(26.3) | 221(26.8) | 9(18.8) | | 53(32.1) | 177(25.0) | | 61(31.0) | 169(25.0) | |
| 32–36 | 138(15.8) | 133(16.1) | 5(10.4) | | 48(29.1) | 90(12.7) | | 38(19.3) | 100(14.8) | |
| ≥36 | 77(8.9) | 76(9.2) | 1(2.1) | | 19(11.5) | 58(8.2) | | 18(9.1) | 59(8.7) | |
| **Neutralizing Antibody** | | | | 0.288 | | | 0.017* | | | <0.001* |
| + | 390(44.7) | 365(44.2) | 25(52.1) | | 60(36.4) | 330(46.6) | | 66(33.5) | 324(47.9) | |
| - | 483(55.3) | 460(55.8) | 23(47.9) | | 105(63.6) | 378(53.4) | | 131(66.5) | 352(52.1) | |
| **SARS-CoV-2 IgG** | | | | 0.706 | | | <0.001* | | | 0.012* |
| + | 405(46.4) | 384(46.5) | 21(43.8) | | 56(33.9) | 349(49.3) | | 76(38.6) | 329(48.7) | |
| - | 468(53.6) | 441(53.5) | 27(56.3) | | 109(66.1) | 359(50.7) | | 121(61.4) | 347(51.3) | |

was a significant difference between the neutralizing antibody positive group and neutralizing antibody negative group in terms of vaccination doses ($P < 0.001$) and time since vaccination ($P < 0.001$). There was a significant difference between IgG antibody negative and positive groups in terms of age ($P = 0.025$), adverse events ($P = 0.027$), vaccination doses ($P < 0.001$), and time since vaccination ($P < 0.001$). Among pregnant women who took iron, there was a significant difference between the negative and positive groups of neutralizing antibody, as well as the negative and positive groups of IgG antibody, in terms of vaccination doses ($P < 0.001$). In pregnant women taking DHA, there was a significant difference between IgG antibody negative and positive groups in terms of gravidity ($P = 0.040$), vaccination doses ($P < 0.001$), and time since vaccination ($P < 0.001$). There was a statistically significant difference between the neutralizing antibody positive group and neutralizing antibody negative group in terms of vaccination doses ($P < 0.001$) and time since vaccination ($P = 0.009$) (Data presented in S1-S3 Tables in S1 File)

### 3.2 Influence of nutritional supplements on antibody levels

To determine the effect of nutritional supplements on antibody levels, multivariable logistic regression analysis was performed. After adjustment for confounding factors such as age, education, vaccine doses, and time since vaccination, we found that there was no significant difference in the positive rates of SARS-CoV-2 neutralizing antibody and IgG antibody according to whether participants took the three nutritional supplements ($P > 0.05$). Compared with pregnant women who did not take folic acid, the odds ratios (ORs) for the presence of SARS-CoV-

**Table 3. Multivariable logistic regression analysis of Nutritional supplements and antibody positivity.**

| Characteristics | P | Neutralizing Antibody OR (95%CI) | P | IgG OR (95%CI) |
|---|---|---|---|---|
| **Folic acid** [a] | | | | |
| No take | | 1.00 | | 1.00 |
| Take | 0.255 | 0.67(0.34–1.32) | 0.547 | 1.24(0.60–2.55) |
| **Iron supplements** [b] | | | | |
| No take | | 1.00 | | 1.00 |
| Take | 0.465 | 1.16(0.77–1.76) | 0.931 | 0.98(0.64–1.49) |
| **DHA** [c] | | | | |
| No take | | 1.00 | | 1.00 |
| Take | 0.085 | 0.71(0.49–1.04) | 0.801 | 0.95(0.65–1.38) |

[a] Model 1: adjusted for age, BMI, education, gravidity, parity, adverse events, inoculation durations, doses of vaccination.

[b] Model 2: adjusted for age, BMI, smoking, number of exercises per week, adverse events, inoculation durations, doses of vaccination.

[c] Model 3: adjusted for age, BMI, smoking, number of exercises per week, adverse events, inoculation durations, doses of vaccination.

2 NAb and IgG antibody in pregnant women who took folic acid were 0.67 (*P* = 0.255; 95% CI, 0.34–1.32) and 1.24 (*P* = 0.547; 95% CI, 0.60–2.55), respectively. Compared with pregnant women who did not take iron supplements, the ORs for the presence of NAb and IgG antibody in pregnant women who took iron supplements were 1.16(*P* = 0.465; 95% CI, 0.77–1.76) and 0.98 (*P* = 0.931; 95% CI, 0.64–1.49), respectively. Similarly, the ORs for NAb and IgG antibody were 0.71 (*P* = 0.085; 95% CI, 0.49–1.04) and 0.95 (*P* = 0.801; 95% CI, 0.65–1.38) in pregnant women who took DHA compared with those who did not (Table 3).

## 4 Discussion

The present study showed no association between supplementation during pregnancy and antibody levels in pregnant women who received inactivated SARS-CoV-2 vaccines.

During the COVID-19 pandemic, vaccination has proven to be the most effective control measure in establishing an immune barrier and reducing the rate of severe disease. Although pregnant women were not included in the clinical trials of COVID-19 vaccines before market release, they face higher severity and mortality rates once infected. Therefore, there is a greater need for pregnant women to get vaccinated in order to protect themselves and their fetuses. With more attention paid to pregnant women's health and nutrient intake, nutritional supplements have been ubiquitously used during pregnancy. Because the development and maintenance of immune cells that support vaccine response depend on a sufficient supply of nutrients, the role of nutritional supplements in immune response to vaccines has drawn increasing attention.

Clinical studies show that low plasma DHA levels during fetal development are associated with cognitive and behavioral disorders [23]. The brain needs DHA supplementation during development of the nervous system. DHA supplementation during pregnancy can increase fetal birth weight, reduce the risk of preterm birth and low birth weight, and also have a positive effect on infant attention regulation [24, 25]. Thus it is necessary to pay attention to DHA intake during pregnancy. In a double-blind, placebo-controlled, randomized trial, children of mothers who took DHA supplements during pregnancy were found to have better problem-solving abilities at age 9 y than those who did not [26]. Taking DHA supplements also improves the health of pregnant women. A randomized controlled trial of low-income African American women showed that pregnant women who received DHA supplementation

reported lower levels of stress and had lower levels of stress hormones [27]. The results of this research indicate that DHA intake is not associated with pregnant women's antibody levels.

Folic acid is essential for DNA and protein synthesis and also plays a crucial role in adaptive immune responses [28]. Pregnancy is a common cause of folic acid deficiency, which can lead to pregnancy complications and affect the growth and development of children, including pregnancy-induced hypertension, increased risk of miscarriage, neural tube defects, cleft lip and palate. Studies have shown that folic acid supplementation during pregnancy may help reduce the risk of prenatal depression and have beneficial effects on fertility [29, 30]. The Centers for Disease Control and Prevention in the USA recommends that women of childbearing age supplement with 400 μg/d of folic acid daily, beginning at least four weeks before conception and continuing through the first three months of pregnancy. In terms of vaccine immune response, taking folic acid during pregnancy increases the persistence of hepatitis B surface antibodies in the fetus [31]. Our study showed that taking folic acid is not associated with pregnant women's antibody levels. This is because the aforementioned study explored the relationship between folic acid supplements during pregnancy and the immune persistence of antibodies in the fetus, which differs from the focus of our research. Other confounding factors in this study, such as vaccine dosage and income level, may also contribute to this result. In a cluster randomized clinical trial conducted in Niger, the effect of prenatal nutritional supplements on immune responses to oral attenuated rotavirus vaccine was assessed. The results showed that in this low-income country, the types of prenatal nutritional supplements have no effect on immune reactions [32].

Iron, an essential nutrient, plays a significant role in the immune system [33]. Every cell and organ system in the human body requires iron for normal development and metabolic function. Iron deficiency often causes anemia during pregnancy, severely threatening the health of pregnant women and fetuses and increasing the incidence of adverse pregnancy outcomes. A meta-analysis showed a dose-response relationship between iron and birth weight, where before reaching a dose of 66mg, every additional 10mg/day of iron supplementation can increase birth weight by 15.1g, reduce the risk of low birth weight by 3%, and decrease the rate of anemia by 12% [34]. In a birth cohort study, anemia and iron deficiency during babies' vaccination were found to predict a decrease in response to vaccines against diphtheria, whooping cough, and pneumococcus. Iron supplementation increases antibody affinity and seroconversion of the measles vaccine [35]. Our study showed that taking iron supplements does not affect antibody levels in pregnant women. This may be because the efficacies of various vaccines differ across populations, age groups, and environmental conditions.

Iron, folic acid and DHA are important nutrient supplements during pregnancy. A cross-sectional study conducted in China found that the use rate of folic acid was the highest (81.7%) and the most stable among various nutrient supplements [36]. Women of childbearing age and pregnant women are at risk of developing anemia, and iron deficiency is the most recognized risk factor for this condition. Pregnancy poses a significant risk of iron deficiency for women. Iron requirements are greatly increased compared to the non-pregnant state. Adequate iron during pregnancy leads to better pregnancy outcomes for both mother and child. During pregnancy, DHA is transferred in large quantities to the fetus through the placenta. The maternal DHA nutritional status directly affects the fetal DHA nutritional status, which in turn affects fetal development.

This research is a cross-sectional study based on real-world data, differing from the aforementioned studies in terms of research design. Furthermore, differences in research subjects and vaccines can also cause different results. For example, the mechanisms of vaccines may vary between the immune systems of fetuses and pregnant women, thereby producing inconsistent immune responses. Finally, readjusting an established immune system may require

additional intervention beyond nutritional supplements, and the most effective supplement has not been identified.

The strength of this study is that it is the first to examine the influence of nutrient supplements taken during pregnancy on the antibody level produced by pregnant women who have received COVID-19 vaccines. The finding of this study provides a certain degree of reference for pregnant women in this regard. We acknowledge some limitations in present study. Because the data on supplements taken by pregnant women were based on self-report, there may have been some recall bias. Another drawback is the lack of survey information on nutrient intake among pregnant women, which could lead to imprecise results. In the future, we may consider measuring the baseline levels of nutritional supplements in the first trimester and evaluating the effects of nutrient supplements such as vitamin D, calcium, and zinc on the immune levels of pregnant women, to provide more effective recommendations on nutrient intake for pregnant women.

## 5 Conclusions

Nutritional supplementation with folic acid, iron, and DHA during pregnancy was not associated with antibody levels in pregnant women who received inactivated SARS-CoV-2 vaccines.

## Supporting information

**S1 File. Demographic characteristics of the participants taking folic acid, taking iron supplements and taking DHA.**
(DOCX)

**S1 Data.**
(XLSX)

## Acknowledgments

We thank all pregnant women who took part in this survey and the health care workers who helped us.

## Author Contributions

**Conceptualization:** Wenhan Yang.

**Data curation:** Qingsong Chen.

**Investigation:** Xi Zhang.

**Methodology:** Xue Han.

**Project administration:** Wenhan Yang.

**Software:** Yajie Gong.

**Supervision:** Qingsong Chen.

**Validation:** Baolan Chen.

**Writing – original draft:** Xi Zhang.

**Writing – review & editing:** Xi Fu.

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
