## [Decision Letter · Decision Letter 0]

7 Aug 2023

PONE-D-23-21116Effects of nutritional supplements on antibody levels in pregnant women vaccinated with inactivated SARS-CoV-2 vaccinesPLOS ONE

Dear Dr. CHEN,

Thank you for submitting your manuscript to PLOS ONE. After careful consideration, we feel that it has merit but does not fully meet PLOS ONE’s publication criteria as it currently stands. Therefore, we invite you to submit a revised version of the manuscript that addresses the points raised during the review process.

We look forward to receiving your revised manuscript.

Kind regards,

Gang Qin, PhD, MD

Academic Editor

PLOS ONE

Journal Requirements:

Additional Editor Comments:

The reviewers raised several concerns on your original submission. Please make major revision according to their suggestions.

Reviewers' comments:

Reviewer's Responses to Questions

**Comments to the Author**

1. Is the manuscript technically sound, and do the data support the conclusions?

Reviewer #1: No

Reviewer #2: Yes

2. Has the statistical analysis been performed appropriately and rigorously? 

Reviewer #1: No

Reviewer #2: Yes

3. Have the authors made all data underlying the findings in their manuscript fully available?

Reviewer #1: Yes

Reviewer #2: Yes

4. Is the manuscript presented in an intelligible fashion and written in standard English?

Reviewer #1: No

Reviewer #2: Yes

5. Review Comments to the Author

Reviewer #1: This manuscript presents a study that enrolled 873 pregnant women to investigate their general demographic characteristics and the use of nutritional supplements, specifically in relation to the serum antibody levels induced by inactivated SARS-CoV-2 vaccines. While the article attempts to explore the association between nutritional supplements and immune effects of these vaccines during pregnancy, the research significance of this relationship is not clearly discussed. The introduction of the article dedicates considerable space to emphasizing the significance of nutritional supplements for pregnant women, but only briefly mentions the immunomodulatory effects of nutrients in the context of COVID-19 susceptibility. This lack of clarity in the study's objective left me uncertain about the authors' intentions. Overall, the manuscript fails to demonstrate the advantages and necessity of the study, resulting in a lack of novel and significant results. As a result, I do not recommend this manuscript for publication in PLOS ONE.

Major concerns:

1. The meaning of the reported P values in Table 1 and Table 3 is unclear. For example, if the P value in line "Age group" (Table 1, 0.054) indicates that there are no significant differences between the folic acid group and non-folic acid group in terms of age, then how should we interpret the P value in line "Doses of vaccination" (Table 3, <0.001)? It suggests that there is a significant difference between the neutralizing antibody positive group and neutralizing antibody negative group in terms of vaccination doses. However, in lines 201-203, the authors conclude that "For pregnant women who took folic acid, the positive rate of SARS-CoV-2 neutralizing antibody varied significantly with vaccine doses (P < 0.001) and time since vaccination (P < 0.001)." Such inconsistencies are present throughout the article. If the P values have different implications in different tables, the authors should provide clear explanations in the manuscript.

2. In lines 186-187, the authors state "We found significant differences in gravidity (P = 0.002), parity (P = 0.012), and education (P = 0.008) according to whether the pregnant women took folic acid." The intended meaning of this statement is unclear. Similar sentences also appear on lines 190-193.

3. In Table 6, the authors adjusted for variables such as age, BMI, education, number of exercises per week, adverse events, inoculation durations, and doses of vaccination. The rationale behind selecting these variables and not others is not explained. Additionally, the manuscript lacks univariate analysis of these variables in relation to antibody levels.

Reviewer #2: This study aimed to examine the influence of dietary supplements on antibody levels in pregnant women vaccinated with inactivated SARS-CoV-2 vaccine. Because it is a cross-sectional study, it is better to use “influence” than “effect”, which may imply a causal association. Overall, this study is well written and seems to add to the literature timely. There are some comments that can be considered for further publication.

Page 5, add citation for “Pregnant women with iron deficiency anemia are more susceptible to SARS-CoV-2 infection.”

Page 5, in the last paragraph of the introduction, please show how this study add to the literature.

Page 5, add a flowchart for the study participants

Page 6, in the study variable section, try to make clear that which variables are outcome variables, exposure variables, and covariates, respective.

Page 7, justify why fisher test are needed.

Page 12, line 200, authors further analyzed the general demographic characteristics of the pregnant women taking the three nutritional supplements separately. Tables 3-5 can be combined to one Table. Also, these tables seem not quite contribute well to the study aim, which can be considered to be the supplemental Table.

Page 20, page 283, it is better to use "multivariable analysis", rather than "multivariate analysis".

Page 21, Table 6 is the final model to answer the study aim. In bivariate analyses, the associations between exposure variables (e.g., iron supplements and DHA) and outcome variables seems to be statistically significant but the associations disappeared in the adjusted models. Any reasons? For each model based on three separate exposure variables, whether authors adjusted for different confounding factors (e.g., see Table 1)? What is the definition of the confounder? Also, are there any effect modifiers? In the statistic analysis section, authors can provide more details.

Page 21, in discussion section, any previous studies in China or worldwide have been done? Are their findings consistent with the present study? Try to add some novelty/strength of this study.

6. PLOS authors have the option to publish the peer review history of their article (what does this mean?). If published, this will include your full peer review and any attached files.

Reviewer #1: No

Reviewer #2: No

---

## [Author Response · Author response to Decision Letter 0]

24 Aug 2023

Aug 13, 2023

Dear Editor,

Thank you for providing us this opportunity to revise our manuscript entitled “Effects of nutritional supplements on antibody levels in pregnant women vaccinated with inactivated SARS-CoV-2 vaccines” (Manuscript ID: PONE-D-23-21116). On behalf of my colleagues, I am submitting our revised manuscript. We appreciate the reviewers’ and editor’s positive and insightful comments. We have carefully considered all the comments and revised the manuscript accordingly. In order to facilitate the review process, we provided a point-by-point response to each of the comments. The precise page and line in the revised manuscript where each change was made in response to the comments were provided as well. 

We appreciated for Editors and Reviewers’ warm work earnestly. We have tried our best to improve and revise the manuscript. We hope that the manuscript is now acceptable. Should you have any additional requests or questions, please do not hesitate to contact me. 

RESPONSES TO THE REVIEWER#1’ S COMMENTS

Comment 1: The meaning of the reported P values in Table 1 and Table 3 is unclear. For example, if the P value in line "Age group" (Table 1, 0.054) indicates that there are no significant differences between the folic acid group and non-folic acid group in terms of age, then how should we interpret the P value in line "Doses of vaccination" (Table 3, <0.001)? It suggests that there is a significant difference between the neutralizing antibody positive group and neutralizing antibody negative group in terms of vaccination doses. However, in lines 201-203, the authors conclude that "For pregnant women who took folic acid, the positive rate of SARS-CoV-2 neutralizing antibody varied significantly with vaccine doses (P < 0.001) and time since vaccination (P < 0.001)." Such inconsistencies are present throughout the article. If the P values have different implications in different tables, the authors should provide clear explanations in the manuscript.

Response: Thanks for the reviewer's comment. As you mentioned above，the P value in line "Doses of vaccination" (Table 3, <0.001) suggests that there is a significant difference between the neutralizing antibody positive group and neutralizing antibody negative group in terms of vaccination doses. I'm sorry for not expressing clearly.We have revised (show in line 212-226) as:

For pregnant women who took folic acid, there was a significant difference between the neutralizing antibody positive group and neutralizing antibody negative group in terms of vaccination doses (P < 0.001) and time since vaccination (P < 0.001). There was a significant difference between IgG antibody negative and positive groups in terms of age (P = 0.025), adverse events (P = 0.027), vaccination doses (P < 0.001), and time since vaccination (P < 0.001). Among pregnant women who took iron, there was a significant difference between the negative and positive groups of neutralizing antibody, as well as the negative and positive groups of IgG antibody, in terms of vaccination doses (P < 0.001). In pregnant women taking DHA, there was a significant difference between IgG antibody negative and positive groups in terms of gravidity (P = 0.040), vaccination doses (P < 0.001), and time since vaccination (P < 0.001). There was a statistically significant difference between the neutralizing antibody positive group and neutralizing antibody negative group in terms of vaccination doses (P < 0.001) and time since vaccination (P = 0.009) .

Comment 2: In lines 186-187, the authors state "We found significant differences in gravidity (P = 0.002), parity (P = 0.012), and education (P = 0.008) according to whether the pregnant women took folic acid." The intended meaning of this statement is unclear. Similar sentences also appear on lines 190-193.

Response: Thanks for the reviewer's comment. We have revised (show in line 196-198,200-204) as:

We found that there were statistically significant differences between groups of pregnant women who did or did not take folic acid in terms of gravidity (P = 0.002), parity (P = 0.012), and education (P = 0.008).

There were statistically significant differences between groups of pregnant women who did or did not take iron supplements in terms of smoking (P = 0.008), weekly exercise frequency (P < 0.001), vaccine doses (P < 0.001), time since vaccination (P < 0.001), SARS-CoV-2 neutralizing antibody (P = 0.017), and IgG antibody (P < 0.001) .

Comment 3:In Table 6, the authors adjusted for variables such as age, BMI, education, number of exercises per week, adverse events, inoculation durations, and doses of vaccination. The rationale behind selecting these variables and not others is not explained. Additionally, the manuscript lacks univariate analysis of these variables in relation to antibody levels.

Response: Thanks for the reviewer's comment. We have revised (show in line 167-173) as:

The adjusted factors were determined through the results of univariate analysis and literature review. Model 1 was adjusted for age, BMI, education, gravidity, parity, adverse events, inoculation durations, doses of vaccination. Model 2 was adjusted for age, BMI, smoking, number of exercises per week, adverse events, inoculation durations, doses of vaccination. Model 3 was adjusted for age, BMI, smoking, number of exercises per week, adverse events, inoculation durations, doses of vaccination.

RESPONSES TO THE REVIEWER#2’ S COMMENTS

Comment 4: Page 5, add citation for“Pregnant women with iron deficiency anemia are more susceptible to SARS-CoV-2 infection.”

Response: Thanks for the reviewer's comment.We have added citation for the word.

Comment 5: Page 5, in the last paragraph of the introduction, please show how this study add to the literature.

Response: Thanks for the reviewer's comment.We have revised (show in line 113-115) as:

This study can provide a certain reference for the question of the influence of women taking nutrients during pregnancy on the level of antibody produced by COVID-19 vaccines.

Comment 6: Page 5, add a flowchart for the study participants.

Response: Thanks for the reviewer's comment.We have added a flowchart in page 6.

Comment 7: Page 6, in the study variable section, try to make clear that which variables are outcome variables, exposure variables, and covariates, respective.

Response: Thanks for the reviewer's comment.We have revised (show in line 140-143) as:

The presence of nutrient supplements was the exposure variable, negative or positive antibody level were outcome variables, and age, BMI, parity, education and income level, smoking, number of exercises per week, doses of vaccination, adverse reactions, and time since vaccination were covariates.

Comment 8: Page 7, justify why fisher test are needed.

Response: Thanks for the reviewer's comment.We have revised (show in line 164) as:

When the sample size n < 40 or the expected frequency T < 1 use Fisher's test.

Comment 9: Page 12, line 200, authors further analyzed the general demographic characteristics of the pregnant women taking the three nutritional supplements separately. Tables 3-5 can be combined to one Table. Also, these tables seem not quite contribute well to the study aim, which can be considered to be the supplemental Table.

Response: Thanks for the reviewer's comment.We have revised.

Comment 10: Page 20, page 283, it is better to use "multivariable analysis", rather than "multivariate analysis".

Response: Thanks for the reviewer's comment.We have revised (show in line 230) .

Comment 11: Page 21, Table 6 is the final model to answer the study aim. In bivariate analyses, the associations between exposure variables (e.g., iron supplements and DHA) and outcome variables seems to be statistically significant but the associations disappeared in the adjusted models. Any reasons? For each model based on three separate exposure variables, whether authors adjusted for different confounding factors (e.g., see Table 1)? What is the definition of the confounder? Also, are there any effect modifiers? In the statistic analysis section, authors can provide more details.

Response: Thanks for the reviewer's comment.We have revised (show in line 167-173) as:

The adjusted factors were determined through the results of univariate analysis and literature review. Model 1 was adjusted for age, BMI, education, gravidity, parity, adverse events, inoculation durations, doses of vaccination. Model 2 was adjusted for age, BMI, smoking, number of exercises per week, adverse events, inoculation durations, doses of vaccination. Model 3 was adjusted for age, BMI, smoking, number of exercises per week, adverse events, inoculation durations, doses of vaccination.

Comment 12: Page 21, in discussion section, any previous studies in China or worldwide have been done? Are their findings consistent with the present study? Try to add some novelty/strength of this study.

Response: Thanks for the reviewer's comment.We have revised (show in line 335-338) as:

The strength of this study is that it is the first to examine the influence of nutrient supplements taken during pregnancy on the antibody level produced by pregnant women who have received COVID-19 vaccines. The finding of this study provides a certain degree of reference for pregnant women in this regard.

---

## [Decision Letter · Decision Letter 1]

4 Sep 2023

Influence of nutritional supplements on antibody levels in pregnant women vaccinated with inactivated SARS-CoV-2 vaccines

PONE-D-23-21116R1

Dear Dr. CHEN,

We’re pleased to inform you that your manuscript has been judged scientifically suitable for publication and will be formally accepted for publication once it meets all outstanding technical requirements.

Kind regards,

Gang Qin, PhD, MD

Academic Editor

PLOS ONE

Additional Editor Comments (optional):

The authors have addressed all of the reviewers' concerns.

Reviewers' comments:

Reviewer's Responses to Questions

**Comments to the Author**

1. If the authors have adequately addressed your comments raised in a previous round of review and you feel that this manuscript is now acceptable for publication, you may indicate that here to bypass the “Comments to the Author” section, enter your conflict of interest statement in the “Confidential to Editor” section, and submit your "Accept" recommendation.

Reviewer #2: All comments have been addressed

2. Is the manuscript technically sound, and do the data support the conclusions?

Reviewer #2: Yes

3. Has the statistical analysis been performed appropriately and rigorously? 

Reviewer #2: Yes

4. Have the authors made all data underlying the findings in their manuscript fully available?

Reviewer #2: Yes

5. Is the manuscript presented in an intelligible fashion and written in standard English?

Reviewer #2: Yes

6. Review Comments to the Author

Reviewer #2: Authors have addressed my comments and revised accordingly in the main text. Additionally, revised tables are improved to satisfy the study aims.

7. PLOS authors have the option to publish the peer review history of their article (what does this mean?). If published, this will include your full peer review and any attached files.

Reviewer #2: No

---

## [Editor Report · Acceptance letter]

27 Feb 2024

PONE-D-23-21116R1 

PLOS ONE

Dear Dr. Chen, 

I'm pleased to inform you that your manuscript has been deemed suitable for publication in PLOS ONE. Congratulations! Your manuscript is now being handed over to our production team.

Kind regards, 

on behalf of

Dr. Gang Qin 

Academic Editor

PLOS ONE